# Syngas Production Improvement of Sugarcane Bagasse Conversion Using an Electromagnetic Modified Vacuum Pyrolysis Reactor

**Muhammad Djoni Bustan [1],\*, Sri Haryati [1], Fitri Hadiah [1], Selpiana Selpiana [1] and Adri Huda [2],\*** 

[1] Department of Chemical Engineering, Faculty of Engineering, Universitas Sriwijaya, Indralaya 30662, Sumatera Selatan, Indonesia; haryati_djoni@yahoo.com (S.H.); gifitri@yahoo.com (F.H.); selpiana@ft.unsri.ac.id (S.S.)

[2] Department of Chemistry, Faculty of Mathematics and Science, Universitas Sumatera Utara, Medan 20155, Sumatera Utara, Indonesia

\* Correspondence: djajashanta@yahoo.co.id (M.D.B.); adri.huda28@gmail.com (A.H.); Tel.: +1-774-502-1404 (A.H.)

**Abstract:** The trends and challenges of pyrolysis technology nowadays have shifted to low-temperature pyrolysis, which provides low-cost processes but high-yield conversion, with suitable $H_2$/CO ratios for performing gas-to-liquid technology in the future. The present study has developed a modified vacuum pyrolysis reactor to convert sugarcane bagasse to gas products, including $H_2$, $CO_2$, $CH_4$, and CO in the low-temperature process. The experimental design includes the effects of pyrolysis time, pyrolysis temperature, and applying a current as a function of the electromagnetic field. The result showed that 0.12 ng/μL, 0.85 ng/μL, and 0.31 ng/μL of hydrogen ($H_2$), carbon dioxide ($CO_2$), and carbon monoxide (CO) gases, respectively, started forming in the first 20 min at 210 °C for the pyrolysis temperature, and the gas product accumulated in the increase of pyrolysis time and temperature. In the absence of electromagnetic field, the optimum condition was obtained at 60 min and 290 °C of pyrolysis time and temperature, respectively, in which 20.98, 14.86, 14.56, and 15.78 ng/μL of $H_2$, $CO_2$, $CH_4$, and CO were generated, respectively. However, this condition did not meet the minimum value of Fischer–Tropsch synthesis, since the minimum requirement of the $H_2$/CO ratio is 2. Furthermore, applying the electromagnetic field performed a significant improvement, in which applying current ≥3A improved the gas product to 33.76, 8.71, 18.39, and 7.66 ng/μL of $H_2$, $CO_2$, $CH_4$, and CO, respectively, with an $H_2$/CO ratio above 2. The obtained result showed that applying electric current as an electromagnetic field provides a significant improvement, not only in boosting yield product, but also in performing the standard ratios of $H_2$/CO in the gas–liquid conversion of syngas to liquid hydrocarbon. The result proves that applying an electromagnetic approach could be used as an alternative way to obtain efficiency and as a better process to convert biomass as a future energy source.

**Keywords:** pyrolysis technology; electromagnetic field; biomass; conversion; syngas production

## 1. Introduction

Converting biomass to energy has become one of the ways to enhance the value of biomass, which mostly becomes an environmental problem as biowaste. The use of biomass as the source of new or renewable energy is supported by an enormous amount of research that has successfully converted the biomass to fuel [1,2]. Literature reviews have reported that biomass-based energy has contributed 14% of world energy consumption, and it keeps increasing annually [3,4]. As an agricultural country,

Indonesia has produced an enormous amount of biowaste, since most of the agricultural activity releases biomass as a byproduct. The amount of biomass continuously increases, since the government of Indonesia has committed to stop importing agricultural products, such as rice, corn, and sugar, and has focused on supplying those products from local farms and plantations. To be specific to our area, South Sumatera, Indonesia, is dominated by sugarcane plantations, which supply the national sugar consumption. However, most of the sugarcane plantations do not know how to utilize the bagasse and release it as biowaste or burn it up to reduce its quantity. This fact is not good in both economic and environmental perspectives, since burning the bagasse in barn and field areas potentially release carbon dioxide or carbon monoxide, sources of greenhouse gases. Therefore, utilization or converting bagasse to fuel can effectively solve two main problems: providing an alternative energy source and reducing the amount of biomass, which commonly becomes a problem and has the potential for the biomass to be burned in barn and field areas and release greenhouse gases.

In the literature review, there are several processes particularly used to convert biomass to energy, such as fermentation [5], gasification [6], pyrolysis [7], catalytic gasification and pyrolysis [8,9], etc. Each technology has its own advantages and disadvantages. For example, the fermentation process provides a clean technology with low-energy consumption, but it takes a long time to process and needs an additional pre-treatment process to convert biomass to carbon-based renewable energy. In terms of pyrolysis, gasification, and catalytic gasification and pyrolysis, these processes offer a promising approach, since they have a short process in converting biomass to liquid and gas phases energy, which is relatively simple and cheap, and provides a high-yield product [9]. However, these technologies require high-temperature processes, in which the cost to produce energy to convert the biomass is not economically suitable for the amount of energy produced. Nowadays, trends in biomass pyrolysis to produce syngas have shifted to develop a low-temperature-based pyrolysis and gasification process by merging and integrating the pyrolysis process with the other technology [10]. Several cracking catalysts have been also applied [9], but most of catalysts only enhance the gas yield and shorten the pyrolysis time.

The present study tries to develop a new basic concept of the pyrolysis process by modifying a conventional pyrolysis reactor using an electromagnetic field. The main idea of using an electromagnetic field is to reduce the pyrolysis temperature by enriching the pyrolysis environment, using applied currents resulting in low-temperature cracking. It is believed that the applied current initiates electron flows in the system and helps to crack the biomass from a complex molecule to its derivatives or simpler molecules, using a reduction pathway. It is similar when we apply a catalyst, which acts as electron support to help in cracking biomass. As a comparison, a series of temperatures were applied to compare the effectiveness of ways to enhance the yield product. We strongly believe that this will become the first study about how an electric field, as the function of an electromagnetic field, affects gas production. Furthermore, the specific purpose of this study is to investigate the feasibility of the modified, electromagnetic-assisted vacuum pyrolysis reactor to boost the early production of gas in a low-temperature reaction, as the product of biomass conversion. The output of this study is proposing the alternative pyrolysis process using a simple pyrolysis reactor, which provides high-yield product with low temperature and energy demands. This study is not the final step in finding a suitable process to convert biowaste in general, or sugarcane bagasse specifically, but the result could be valuable information to determine the next steps.

## 2. Materials and Methods

### 2.1. Preparation of Biomass Sample

The bagasse sample used in this study was obtained from a sugarcane plantation in South Sumatera, Indonesia. The preparation of the biomass sample was begun by washing the sugarcane bagasse, using clean water several times to separate the impurities, and then the sample was dried under sunlight for 48 h. The dried bagasse was then cut to 5–8 cm sections and chopped using a

crusher to obtain 1–2 mm bagasse particles. The obtained bagasse was rewashed using distilled water several times, and dried in the oven for 24 h at 105 °C to evaporate the water so there was less than 10% moisture by weight. After being dried, the sample was ready to be used. In the case where the sample was being stored, the dried sample was placed in a plastic bag and vacuumed to avoid moisture adsorption, and stored in a freezer with a temperature of −20 °C until the pyrolysis experiment was conducted. In addition, the elemental composition of the sample was characterized using the proximate and ultimate analysis, using the American Society of Testing and Materials (ASTM) standards and International Standardization Organization (ISO) as reference [11,12]. The result and detailed method used to characterize the sample were presented in Table 1.

**Table 1.** Properties of bagasse sugarcane.

| Components of Proximate Analysis | Standard Method | Value [a] | Components of Ultimate Analysis | Standard Method | Value [b] |
|---|---|---|---|---|---|
| Moisture (%) | ASTM E 871-82 | 3.01 | Carbon/C (%) | ASTM E 777 | 45.01 |
| Volatile (%) | ASTM E 872-82 | 72.66 | Oxygen/O (%) | By difference | 41.73 |
| Fixed Carbon (%) | ASTM D 3172-02 | 12.89 | Hydrogen/H (%) | ASTM E 777 | 5.59 |
| Ash (%) | ASTM D1102-84 | 8.98 | Nitrogen/N (%) | ASTM E 778 | 0.29 |
| | | | Sulphur/S (%) | ASTM # 775 | 1.08 |

[a] Dry basis; [b] dry ash-free.

### 2.2. Bagasse Pyrolysis Using Modified-Vacuum Pyrolysis Reactor

The pyrolysis of bagasse was conducted in a stainless-steel fix batch reactor (lab-scale reactor), using a modified-vacuum pyrolysis reactor designed by the Chemical Engineering Department of the Graduate School, University of Sriwijaya (Figure 1). A control panel was set up to adjust and control the reaction temperature, and a set of vacuum pumps was used to create both vacuum and oxygen-free environments for pyrolysis cracking (in range of 5–20 kPa). An approximately 100 g bagasse sample was placed on the biomass chamber in the pyrolysis reactor (CC-01) and heated up to 210 °C for the reaction temperature. To study the early formation of syngas production, the gas collection as a function of reaction time at 20, 30, 40, 50, and 60 min was done by carefully collecting the gas product in liquid storage (LS-01). After finding the optimum pyrolysis time condition, shown by the highest syngas product, two approaches were used to optimize the conversion process. The first approach was conducted by setting a series of pyrolysis temperatures, started from 210 °C, 230 °C, 250 °C, 270 °C, and 290 °C. The second method was performed by applying an electromagnetic field as a function of applied current, starting from 1, 2, 3, 4, and 5 Ampere. In this study, the gas-based products became the main products being harvested during the experiment. Therefore, the amount of char, liquid, and solid-based product produced during the experiment was not monitored. Furthermore, the generated gas in all experiments was analyzed using gas chromatography (Agilent 7890A) equipped with a thermal conductivity detector (TCD), with Argon gas as a carrier gas. The sample of generated gas was obtained from gas storage (GS-01), using a sealed gas sampling equipment.

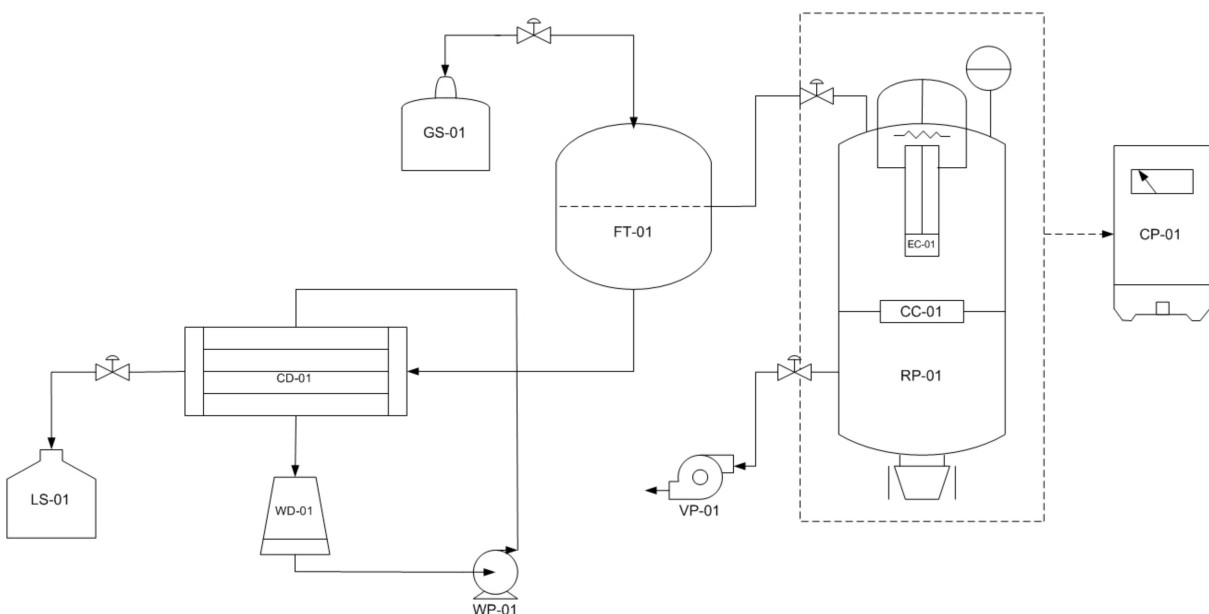

**Figure 1.** Modified vacuum pyrolysis diagram for converting bagasse to syngas. CC-01: biomass chamber; CP-01: control panel; CD-01: condenser; EC-01: electromagnetic induction source; FT-01: flash tank; GS-01: gas storage; LS-01: liquid storage; RP-01: pyrolysis reactor; VP-01: vacuum pump; WD-01: water drum; and WP-01: centrifugal pump.

## 3. Results and Discussion

The study of the effect of pyrolysis time was focused on investigating the early formation of gas as the result of biomass conversion in the absence of oxygen (vacuum). Figure 2 shows that in the first 20 min of resident time, 0.12 ng/μL, 0.85 ng/μL, and 0.31 ng/μL of hydrogen ($H_2$), carbon dioxide ($CO_2$), and carbon monoxide (CO) gases, respectively, started to form, indicating that the conversion process of bagasse had started and that the amount of gas continuously accumulated until 60 min of pyrolysis time, which was the maximum time of experiment used. All the experiments were conducted with 210 °C as the starting pyrolysis temperature. The result was correlated with the work reported by Hlavsová [13], who reported that the early formation of syngas begins from the temperature of 170 °C. The results proved that the reactor could perform the conversion in low temperature process. The reactor performance testing was necessary, since the reactor was built by us. However, there was no methane ($CH_4$) found in all resident times, indicating the obtained system (temperature and pyrolysis time) cannot support converting the organic-based intermediate product to methane, due to the lack of energy generated; the energy was reported at around 166–250 kJ/mol [9]. The absence of methane gas as product of a pyrolysis was also because of the low concentration of CO and $H_2$ as the main ingredients of methane gas in the system [14]. Meanwhile, the high concentration of $CO_2$ cannot initiate the methanation reaction, due to the low concentration of $H_2$. The proposed methanation reaction through CO and $CO_2$ with $H_2$ are shown in Equations (1) and (2):

$$CO + 3H_2 \rightarrow CH_4 + H_2O \tag{1}$$

$$CO_2 + 4H_2 \rightarrow CH_4 + 2H_2O \tag{2}$$

To be more specific in hydrogen gas evolution, all the resident times were generated a relatively similar amount of hydrogen gas as a product, meaning there was no optimal time condition to generate hydrogen gas. Interestingly, the most significant difference was the amount of generated $CO_2$ and CO, which were simultaneously increased by adding pyrolysis time. The most possible reason is because of the formation of $CO_2$ and CO is the first stage of biomass conversion. This is correlated with the

work of Jaffar [14], who reported that the pyrolysis biomass was initiated by producing a suite of gases, including $CO_2$ and CO, and continued by the methanation reaction. From our perspective, there will be a combination of reduction and oxidation reactions in the first step of biomass conversion, indicated by the early formation of hydrogen (reduction reaction) and continued by the formation of $CO_2$ and CO as products of the oxidation process (Equations (3) and (4)).

$$C_nH_mO_p \rightarrow C_{n-x}H_{m-y} + H_2 + CO_2 + C \quad \Delta H_r^0 \geq 0 \text{ kJ/mol} \tag{3}$$

$$C + CO_2 \rightarrow 2CO \quad \Delta H_r^0 \text{ 173 kJ/mol} \tag{4}$$

The study was continued by investigating the effect of pyrolysis temperature on the increase of sugarcane bagasse conversion. The applied temperature was chosen below 300 °C, to ensure that all pyrolysis temperatures were below the common and commercial pyrolysis temperature (400–1000 °C) [15–17]. The result was shown in Figure 3. The highest yield was found at the pyrolysis temperature of 290 °C, in which the amount of $H_2$, $CO_2$, $CH_4$, and CO gases was 20.98, 14.86, 14.56, and 15.78 ng/μL, respectively. On the other hand, the lowest yield was obtained at 210 °C as a control, with the amount of $H_2$, $CO_2$, $CH_4$, and CO gases at 0.12, 1.78, 0.00, and 0.83 ng/μL, respectively. To be more detailed, there was a slight increase of gas CO, $CO_2$, and $CH_4$ in the first step of temperature enhancement to 230 °C (from 210 °C as the first approach/control). The number of CO and $CO_2$ were improved to 5 and 12 times, respectively, compared to the system at 210 °C. At 230 °C, $CH_4$ was produced for the first time (6.78 ng/μL), indicating that 230 °C had become the initial temperature of methanation reaction. Nevertheless, there was no hydrogen gas found at 230 °C, indicating that the partially generated hydrogen was used to initiate the formation of $CH_4$ through the methanation process (see Equations (1) and (2)) [13]; however, the amount of $H_2$ continuously formed after 230 °C and slightly improved after 250 °C, indicating that thermal cracking continued to convert the biomass to produce $H_2$ gas [13]. However, the amount of CO, $CO_2$, and $CH_4$ were decreased after the temperature of 270 °C, because the formation of char initiated the biochemical component of released gases and carbon [18].

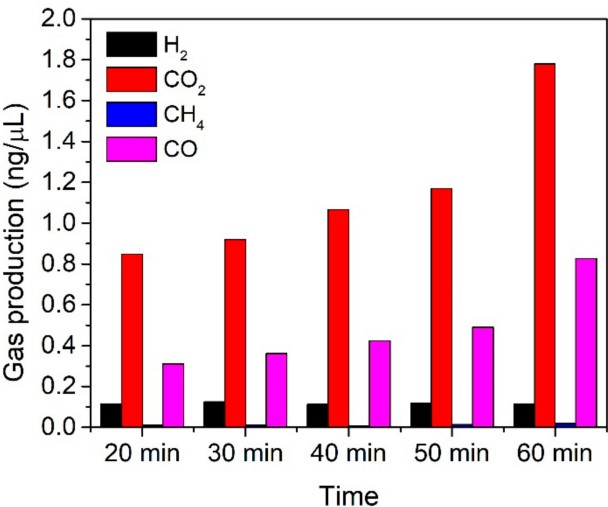

**Figure 2.** The effect of pyrolysis time in gas production. Temperature: 210 °C, with vacuum pressure in the range of 5–20 kPa.

The result of yield production was further utilized to see the $H_2$/CO ratio, in order to see the potential for the conditions to produce liquid hydrocarbon. The results showed that the high yield at 290 °C did not produce the highest $H_2$/CO ratio; the highest $H_2$/CO ratio, which was 1.54, was obtained at a pyrolysis temperature of 250 °C, followed by the system at temperatures of 290 °C, 270 °C, and 230 °C (with $H_2$/CO ratios of 1.33, 1.33, and 1.18, respectively). However, the whole system

has relatively low $H_2/CO$ to perform liquid hydrocarbon conversion, because the minimum ratio required is 2–3 [19]. Considering $H_2/CO$ ratio and yield of gas produced, the temperature of 290 °C was considered as the optimum condition to convert biomass to gas-based product.

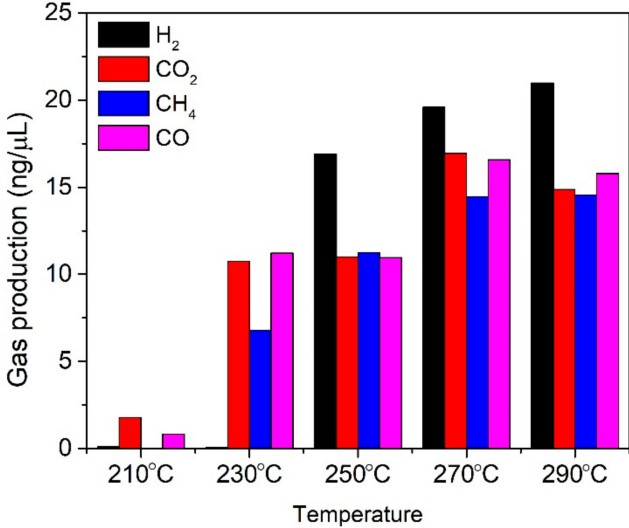

**Figure 3.** Total yield syngas production at 60 min of pyrolysis time as a function of pyrolysis temperature.

The study was continued by investigating the effect of applying an electromagnetic field to biomass conversion. The results shown in Figure 4 show that applying the electromagnetic field strongly improved the hydrogen generation compared to control, in which injecting of 5A currents provide the highest yield (33.76 ng/μL), followed by 4A (30.89 ng/μL), 3A (17.95 ng/μL), 2A (8.99 ng/μL), 1A (0.91 ng/μL), and the control as the lowest hydrogen generation (0.11 ng/μL). The slight improvement of hydrogen generation produced the enhancement of $H_2/CO$ ratio (Table 2), in which several conditions with an applied current ≥3A had $H_2/CO$ ratios greater than or equal to 2, which was suitable to convert syngas to liquid hydrocarbon. Furthermore, the results showed that applying an electromagnetic field to the system enhanced carbon conversion efficiency. After applying 2A, the amount of $CO_2$, $CH_4$, and CO improved from 1.78 ng/μL, 0.00 ng/μL, and 0.82 ng/μL, respectively, to 8.21 ng/μL, 7.03 ng/μL, and 9.99 ng/μL, respectively. The amount of methane also seemed to improve after increasing of applied current. The most possible reason was that the applied current facilitated improving the pyrolysis temperature, due to the magnetic field and different heat transfer efficiencies of the induction media. Furthermore, the electromagnetic field facilitated polarity in the lignocellulose media or carbon-based product released in the first conversion ($CO_2$, and CO), due to the dipole rotation on an atomic scale, resulting in the improvement of the heating rate of biomass conversion and initiating further methanation reaction [20–22]. It was shown by the enhancement of methane that applying a 5A current consumed $CO_2$ and CO, demonstrated by a decreased amount of $CO_2$ and CO comparing to the condition of applied 4A current.

**Table 2.** The ratio of $H_2/CO$ in the pyrolysis system modified with electromagnetic field.

| Applied Current (A) | Temperature (°C) | Pyrolysis Time (Minutes) | $H_2/CO$ Ratio |
|---|---|---|---|
| 0 | | | 0.13 |
| 1 | | | 0.66 |
| 2 | 210 | 60 | 0.89 |
| 3 | | | 1.98 |
| 4 | | | 3.18 |
| 5 | | | 4.40 |

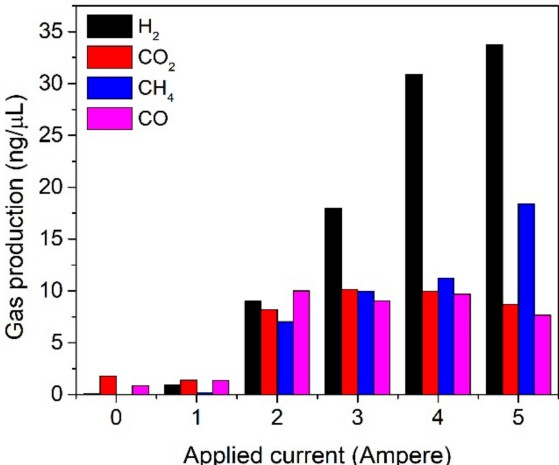

**Figure 4.** Syngas production improvement in the increasing of an applied electromagnetic current. Pyrolysis time and temperature were controlled at 60 min and 210 °C, respectively (control).

## 4. Conclusions

Early formation of CO, $CO_2$, and $H_2$ observed at 210 °C indicated that biomass can be transformed at low temperatures under 60 min pyrolysis time. However, the initiated product cannot be further utilized, due to the low number of generated products. The study provides two solutions to enhance the conversion process: using temperature and electromagnetic pathways. In the temperature approach, all the pyrolysis gas product slightly improved by increasing the temperature above 210 °C. The methanation process seemed to start at 230 °C. However, the $H_2/CO$ ratios in the produced gases were lower, to fulfill the standard requirement for further processes to convert them to liquid-based hydrocarbon. The second approach, using an electromagnetic field, showed a better result, where the system effectively converted biomass to gas-based products under a low temperature (210 °C). Applying an electromagnetic field could also initiate methanation by applying a 2A current. Furthermore, applying ≥3A of current potentially converts gas product to liquid-based hydrocarbon through the Fischer–Tropsch synthesis pathway, since the $H_2/CO$ ratio was more than or equal to 2. The result showed that applying electromagnetic field potentially converted biomass more effectively than increasing the temperature of pyrolysis.

**Author Contributions:** Conceptualization and supervision, M.D.B. and S.H.; data curation, S.S.; funding acquisition, M.D.B.; investigation, F.H. and A.H.; methodology, M.D.B., S.H., and A.H. All authors have read and agreed to the published version of the manuscript.

**Funding:** This research is fully funded by Universitas Sriwijaya.

**Acknowledgments:** The author acknowledges the Department of Chemical Engineering, University of Sriwijaya for facilitating the research.

**Conflicts of Interest:** The authors declare no conflict of interest.

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
