# Peer review of "Syngas Production Improvement of Sugarcane Bagasse Conversion Using an Electromagnetic Modified Vacuum Pyrolysis Reactor"

_processes, doi:10.3390/pr8020252_

Round 1

Reviewer 1 Report

General remarks:

In the introduction, there is no justification for undertaking research on vacuum pyrolysis combined with an electromagnetic field. The description of the experimental set-up is unclear. What is the role of GS-01 in this system? Parameters that were controlled during pyrolysis should be given. There is no information about the frequencies of electromagnetic field used in experiments. No characteristics of sugarcane bagasse used in experiments is included in the manuscript. Such information would be helpful to explain e.g. no solid residue after pyrolysis. Optimization of process parameters usually is done using e.g. methods of design of experiments. In my opinion, the presented results do not justify that the pyrolysis was done at optimal conditions of time and temperature. line 160 and next: The discussion of the results should be based on the presented results. I did not find information about the results of kinetics of pyrolysis, so the statement that “electromagnetic field also shorten the first step of pyrolysis reaction pathway” is not justified.

Other remarks:

The quality of Figure 1 should be improved. The unit of syngas production in Figure 2, 3 and 4 is unclear; ng/µL means the mass of syngas component (H2, CO2, CO, CH4) per volume of gas? What was the pressure of the syngas? What is the yield of syngas per mass of biomass? Figure 4: The values of “Syngas production” for 0A is different than those presented in Figure 3 for 250°C and 60 min (e.g. 1,70E+01 for H2 in Fig 3, less than 1,00E+00 in Fig.4).

Author Response

Dear Reviewer Here our responses related to your question.

First of all, thank you for your constructive question and suggestion. It is meant to help us improve the quality of our work manuscript. We also have carefully evaluated and improved the paper.

Question: There is no justification for researching vacuum pyrolysis and electromagnetic field

Answer: We have updated our introduction by adding sentences about how we choose vacuum pyrolysis and electromagnetic field. 

Question: Experimental design is unclear including the role of GS-01

Answer: We have updated by adding the function of GS-01 as the place to trap the generated gas before analyzing using gas chromatography.

Question: What is the controlled parameter?

Answer:

1. Temperature (210oC) is set for the first experiment. The choice of the temperature is based on our preliminary experiment which showed that 210oC as the minimum temperature to do the pyrolysis. 

2. Vacuum pressure is set up in the range 5-20 kPa.

3. Pyrolysis time (20-60 minutes)

4. Applied current as of the function of the electromagnetic field. However, the exact frequency was not measured. 

Question: There is no characteristic of sugarcane bagasse

Answer: We have updated our manuscript by adding the result of the proximate and ultimate analysis.

Question: No Solid Residue Data Showed in Manuscript

Answer: Our main parameter investigated during our study is to see if the applied current can enhance the syngas production which is CO2, CO, H2, and CH4. There will residue after the experiment but we do not focus on it.

Question: Optimization of the process is not well presented and explained. 

Answer: In this manuscript, we tried to see the early formation of gas as the product of biomass conversion. However, there is no study of optimization conducted. You will see that there are two approaches used (temperature and applied electromagnetic field), but these two approaches are only to see how every approach improved the conversion rate. 

Question: The unit of syngas production in Figures 2, 3 and 4 is unclear; ng/µL means the mass of syngas component (H2, CO2, CO, CH4) per volume of the gas? What was the pressure of the syngas?

Answer:  The unit of gas production is based on the amount of gas produced per amount of gases taken during the sampling process. Unfortunately, our reactor is not online with gas chromatography, in which the number of gases taken in every sampling time was quite similar but different in the number of gas components. We evaluated the pyrolysis process by conducting sampling every 60 minutes by opening the valve which connected to the sampling chamber. The pressure of gas should be similar to the pressure of the reactor. 

Question: What is the yield of syngas per mass of biomass?

Answer: We could not determine precisely the yield of syngas per mass of biomass since the conversion process is not fully conducted or the conversion process is not finished yet. We have to tell that this study only investigated the early formation of gas released during the biomass conversion. This study was not specifically focused on converting all the biomass to the gas-based product. This is also the reason why char, liquid-based products, did not well count during the study.  

Question: Figure 4: The values of “Syngas production” for 0A is different than those presented in Figure 3 for 250°C and 60 min (e.g. 1,70E+01 for H2 in Fig 3, less than 1,00E+00 in Fig.4).

Answer: Thank you for correcting it. We wrote mistakenly and have updated the data.

We have carefully evaluated our manuscript based on the reviewer's suggestion. 

Reviewer 2 Report

The effects of temperature and reaction time and electromagnetic field on the production of syngas were reported in this letter. The letter is interesting I think.
However, the following point needs to be considered:
1. The language and notations need to be improved in your letter
2. Please explain what optimum pyrolysis how to find it is.
3. Please show the heat of reaction in Eqs.(1)-(3).
4. In line 156, you state that applying electromagnetic field changes the pyrolysis temperature. Please show the evidence if possible.
5. Power supply was required when using electromagnetic field. Can you compare the energy requirement between vacuum pyrolysis with electromagnetic field and thermal pyrolysis, which one is better?

Author Response

Dear Reviewer

Thank you for your question and suggestion.

We have carefully evaluated the manuscript and updated it based on your suggestion. Here some of our response related to your question.

Question: The language and notations need to be improved in your letter

Answer: We have updated our manuscript. We thank you for your suggestion

Question: Please explain what optimum pyrolysis how to find it is.

Answer: We have to say that we did not comprehensively conduct the optimization process. First, we try to investigate the early formation of gas production as a sign of biomass conversion. We see that time is a key parameter, thus the effect of pyrolysis time is conducted. 

After finding a suitable time to analyze it, we try to see if we can boost the product using two approaches which are increasing pyrolysis temperature and applying electromagnetic fields. Each process has a different effect. If we have to say what is the optimum condition,  we can say that 60 minutes, 290oC, and applying 5Ampere of current are the optimum condition to convert biomass to gas. however, we just interpreted the data based on the parameter and did not investigate how each parameter correlated to obtain the optimum condition. 

Questions: Please show the heat of reaction in Eqs. (1)-(3).

Answer: We are sorry that we did not get what you were asking.

Question: Inline 156, you state that applying the electromagnetic field changes the pyrolysis temperature. Please show the evidence if possible.

Answer: We could not show the evidence since our thermocouple showed a similar number of temperatures. we could say that the literature review mentioned that the high yield conversion using the microwave or electromagnetic field was because of the formation of polarity or dipole rotation in the atomic-scale inside of biomass which is caused by the electromagnetic field. 

Question: Power supply was required when using the electromagnetic field. Can you compare the energy requirement between vacuum pyrolysis with electromagnetic field and thermal pyrolysis, which one is better?

Answer: We could say that the electromagnetic field required additional energy to run the system. But in the case of which one is better, we could say that the electromagnetic field possesses a better performance since providing high yield and better H2/CO ratios. 

Best

Authors

Reviewer 3 Report

This manuscript is very poorly written. Besides extremely poor English, the manuscript also lack concise scientific expressions and at times fails to communicate what novelty or new idea and approach authors wish to bring forward. Experimental presented in this manuscript do not relate to conclusions drawn.

Author Response

Dear Reviewer

We thank you for your comment. It is worthy to develop our manuscript much better than before. Moreover, we have updated our manuscript and hopefully, it can fulfill your requirements. We could say that our manuscript is lack of data and interpretation since it is only submitted in Letter Journal or Short Communication than Original Paper/Full Length Paper. The result is still run and developed until now. We just tried to "Announce" what we got so far.

Best

Authors

Reviewer 4 Report

see attached file

Author Response

Dear Reviewer

Thank you for your comments and suggestions. We have evaluated our manuscript and read some papers that you recommend. We have evaluated the content and also English language. Hopefully, it can fulfill your standard.

Thank You

Author

Round 2

Reviewer 1 Report

I have still a remark to the analysis of sugarcane bagasse, because there is no information on the analytical method for biomass composition in the section “Materials and Methods”.

Small remarks:

a) line 612

It is: “Table 1”; is should be “Table 2”

b) The temperature in the table with the H2/CO ratio does not match the data in the Figure 4 and in the text of Conclusions (250 or 210°C?).

Author Response

Dear Reviewer. 

First, we acknowledge for your kindly response and remarks to our prepared manuscript. We have carefully added and revised our manuscript, including the detailed experiment, some editing in the English language, and clearly the interpretation. Furthermore, the detailed response will be addressed below.

Question: I have still a remark to the analysis of sugarcane bagasse because there is no information on the analytical method for biomass composition in the section “Materials and Methods”

Answer: to characterize the biomass composition, our standard method refers to  American Society of Testing and Materials
(ASTM) standards and the International Standardisation Organisation (ISO). The detail of any parameter has been added to the manuscript. 

Question: Please correct the remark in lines 612 and table 2.

Answer: We have corrected the typographical error in lines 612 and changing the temperature in table 2.

Thank you for all the corrections and questions.

Best

All authors

Reviewer 3 Report

This manuscript titled “Syngas production improvement of sugarcane bagasse conversion using an electromagnetic modified vacuum pyrolysis reactor” claims to develop a non-conventional way of sugarcane bagasse conversion to to syngas. The title is attractive and authors boast to discuss novelties, the results which do not correspond to evidence produced in the manuscript. I was provided only the pdf print of revised word version of the manuscript which did not allow me to see what exactly the authors had deleted in the so-called revised manuscript.

In the original manuscript, authors had presented the similar approach. In the revised version, I do not see authors giving any serious consideration to our previous observations. Instead, authors have only tried to deflect the reviewers by changing some sentences here and there without any noticeable improvements in the quality of the manuscript.

The first sentences in the abstract itself does not make any sense “Pyrolysis become one of the most technology used to produce syngas from biomass. The trend and challenge of pyrolysis technology nowadays have shifted to low-temperature pyrolysis which provides low-cost technology but providing high yield conversion and suitable H2/CO ratios for performing gas-to-liquid technology in the future.” As we navigate in the manuscript, it becomes more and more difficult to get any meaningful impression of the research boasted in this “Letter”. In my humble opinion, neither the quality of the scientific content presented nor language can be acceptable in the current form.

Author Response

Dear Reviewer

Thank you for your comments related to our revision manuscript. We do not exactly how the reviewing process if the editor provides the reviewer the original manuscript in Pdf or Word format. I think it is out of our control.

Related to your comments. We believe that each reviewer has their own standard to assess the quality of the manuscript. We thank you for your assessment. It will become valuable things for us to improve our research quality and writing skills. We feel sorry if we cannot satisfy you.

Thank you for all the suggestion and comment

Best,

Authors

Reviewer 4 Report

Authors have successfully responded to my comments. Thus, my reccomendatiion for the revised manuscript is to be accepted for publication

Author Response

Dear Reviewer

Thank you for all your kindly suggestion and questions.

We have revised our English language quality of the manuscript. Hopefully, it fulfils your requirements.

Thank you